# Experimental investigation of flexible eight-port asymmetric fed MIMO antenna with narrow-super-wideband$_{n-s}$ characteristics for future applications including internet of things

**Manish Sharma[1]**, **Selvaraj Praveen Chakkravarthy[2]**, **Sathishkumar Nallusamy[3]**, **Kanhaiya Sharma** [4]\*, **Juliano Katrib[5,6]**, **Rana Gill[7]**

**1** Chitkara University Institute of Engineering and Technology, Chitkara University, Rajpura, Punjab, India, **2** Department of BME, Dr. N.G.P. Institute of Technology, Coimbatore, Tamil Nadu, India, **3** Department of Electronics and Communication Engineering, KPR Institute of Engineering and Technology, Coimbatore, Tamil Nadu, India, **4** Department of Computer Science and Engineering, Symbiosis Institute of Technology, Symbiosis International (Deemed University), Pune, Maharashtra, India, **5** GUST Engineering and Applied Innovation Research Center (GEAR), Gulf University for Science and Technology (GUST), Hawally, Kuwait, **6** Department of Electrical and Computer Engineering, Gulf University for Science and Technology (GUST), Hawally, Kuwait, **7** Department of Electronics and Communication Engineering, University Centre for Research and Development, Chandigarh University, Mohali, Punjab, India

These authors contributed equally to this work.
\* kanhaiya.sharma@sitpune.edu.in; sharmakanhaiya@gmail.com

## Abstract

In this work, an eight-port MIMOn-s antenna with dual-band operation is presented. The proposed work involves several wireless applications with FR1 and FR2 millimeter-wave bands. Also, a need for conformal antenna with capability of reduced SAR can cater the need for applications in flexible electronics. The antenna includes an oval-shaped radiating patch with a circular slot printed on Rogers RT Duroid TM5880 substrate on one plane and rectangular defected-ground-structure on the other side with a dimension of $70.71 \times 70.71$ mm$^2$. The radiating patch and ground are asymmetric-fed producing measured bandwidth of Band 1 = 3.59–3.85 GHz useful for WiMAX application and Band 2 = 8.40–70.0 GHz for application in Industry-Scientific-Medical (ISM)-24.0 GHz, Ultra-wideband (UWB)-24.0 GHz and millimeter-wave 5G-FR2 bands (n257-n263). The property of the substrate thickness = 0.254mm is utilized for bending angles of 15°, 30°, and 45° ensuring no change in -10.0dB impedance bandwidth making it suitable for flexible-electronics applications. The SARn-s is evaluated with antenna placed on tissue (skin, fat, muscle) model at SAR/frequency and found to be ≤ 1.60 W/Kg at key frequency values in the operating bandwidth of interest corresponding to 0.0331 W/Kg at 3.50GHz, 0.0435 W/Kg at 10.0GHz, 0.0131 W/Kg at 15.0GHz, 0.0209 W/Kg at 24.0GHZ, 0.0171 W/Kg at 26.0GHz, 0.0116 W/Kg at 28.0GHz, 0.00334 W/Kg at 38.0GHz, 0.00336 W/Kg at 39.0GHz, 0.00296 W/Kg at 41.0GHz, 0.00149 W/Kg at 47.0GHz and 0.0851 W/Kg at 60.0GHz. The diversity performance is also evaluated with measured ECCn-s < 0.02,

**Data availability statement:** All relevant data are within the paper and its Supporting Information files.

**Funding:** The author(s) received no specific funding for this work.

**Competing interests:** The authors have declared that no competing interests exist.

DGn-s > 9.9998dB, TARCn-s < -8.0dB, and CCLn-s < 0.10b/s/Hz. The measured peak-gain varied between 2.965dBi-16.258dBi with maximum peak-gain of 16.13dBi at 60.0GHz including 2-D radiation patterns at WiMAX and other bands. The multi-band flexible-ability and acceptable SARn-s suggest the proposed well suited for WiMAX and future 5G applications in FR2 bands.

## 1. Introduction

The wireless communication system device has become more compact with the development of electronic technology in the past decade. The antenna embedded within it plays a vital role in the communication between the transmitter and the receiver. The planar antenna has also been developed for several wireless applications such as WiMAX, WLAN, microwave-band, and mm-wave bands [1]. For lower RF band application, the size of the antenna is larger which is comparable to wavelength and vice-versa for higher frequency. Due to the challenge of limited space allocated on the printed circuit board, the need for either a multi-band single-patch antenna or super-wideband antennae is required. This work aims to design the patch antenna on a thinner substrate achieving higher impedance bandwidth. The substrate with thickness of more than 0.50mm are not capable of inheriting conformal characteristics and hence, the thickness of 0.254mm is used in the proposed work which can be easily bent at different required angles which is discussed in further sections of the manuscript.

The literature reports as per the characteristics of the proposed work which is inclusive of super-wideband characteristics, conformal capabilities and the application for on-body applications. The literature do not report all the said characteristics in single research article and hence, the research data base comprises all the papers related to three separate terminology which is discussed further. The proposed work includes multiple-input-multiple-output (MIMO) configuration with the ability for applications in flexible and on-body applications with specific-absorption-rate (SAR) analysis. Hence, the literature focuses on the super-wideband antenna [1–9], the flexible capability of the antenna [10–15], the SAR calculation [16–21], and the multi-port MIMO antenna [22–42].

The super-wideband antenna which maintains a bandwidth ratio of more than 10:1 is reported [1] addressing several patch-designed achieving super-wideband configurations. Hexagonal geometry including reconfigurable characteristics with an antenna dimension of 36mm×36mm achieves a super-wideband bandwidth of 3.37GHz-27.71GHz [2] and the integration of a solar panel with the transparent CPW-fed antenna induces a bandwidth of 2.0GHz-32.0GHz with a bandwidth ratio of 15:1 [3]. A Guitar-shaped patch with defected-ground-structure (DGS) and embedded band-stop filters not only eliminates three interfering bands but also results in a bandwidth of 2.76GHz-39.53GHz and Utilizing a Mickey-shaped patch with an identical slot and elliptical ground with elliptical-DGS shows the property of super wideband characteristics with a bandwidth ratio of 38.9:1 [4–7]. A super wideband antenna with a

bandwidth of 2.34GHz-20.0GHz uses a cross-elliptical radiating patch with an attached stub integrating a narrow Bluetooth band [8,9]. As discussed, the literature also includes flexible-characteristics antenna with specific-absorption-rate (SAR) analysis [10–20]. A flexible antenna utilizing conformal-substrate is used to design for application within 4.00GHz-6.00GHz [10] with a compact size of 20mm × 14mm. A UWB-monopole antenna with a Charkha-shaped patch is analyzed for both flexible and SAR applications [11]. Also, the SAR analysis discussed at any frequency value within the operating bandwidth should be limited to 1.60W/Kg. Substrates such as Denim, Cotton, Kapton, polyester, Rogers RTDuroid substrate, etc. are used as flexible substrates for wearable applications with SAR ≤ 1.60W/Kg [39]. A 3-D model with Blood, Fat, Muscle, and Bone layer constitutes to model with an antenna placed above for SAR analysis [15] at frequency values corresponding to 1.80GHz, 2.40GHz, 5.0GHz, and 8.90GHz respectively. The applications of the wearable antenna are useful in transferring the data collected, in diagnosis, and also for the treatment of cancer tissues [16]–[20].

The literature also includes the multi-input-multi-output (MIMO) antenna with two-port [21–23], four-port [24–36] and Eight-port [37–41] configuration. A modified rectangular-patch CPW-fed two-port [21] MIMO antenna with a unique funnel-shaped isolation technique offers an impedance bandwidth of 3.90GHz to more than 150.0GHz. Two-port MIMO antenna with characteristics-mode-theory (CMT) analysis [22–25] helps design the millimeter wave antenna. A palm-tree-shaped patch MIMO antenna with orthogonal orientation comprises of four-port configuration achieving a bandwidth of 23.40GHz-35.0GHz which includes numerous 5G application bands [26]. A hexagon-modified four-port MIMO antenna [27–35], placed in orthogonal sequence is designed for WLAN applications including dual-narrow bands and a combination of a two-square patch fed by corporate feed and two complimentary-split-ring-resonators (CSRRs) form the four-port MIMO antenna with an enhanced gain of 11.0dBi within the operating bandwidth of 26.50GHz-30.40GH. A four radiating patch tree-shaped are placed adjacent-mirrored fashion [36], with two operating bands including WLAN and System and an orthogonal shape geometry of the substrate with inter-connected ground achieves high isolation and operational bandwidth of 3.03GHz-15.33GHz [37–40] The Eight-port MIMO antenna configuration [41,42] shows the utilization of the neutralization line attached between the radiating patch reducing the inter-element interference. A unique placing of an Eight-radiating patch with rectangular DGS-ground [41] includes operational-bandwidth between 23.30GHz-27.60GHz which finds application in 5G-millimeter wave band. The calculation of elliptical patch [43] is reported where the elliptical slots are controlled by RF-MEMS switch to achieve useable bandwidth. Also, the Eight-port MIMO antenna [44–53] are discussed with different combination of arrangements of the radiating patch to achieve diversity. The on-body applications where tissue model is used for SAR analysis [54,55] is needed with electrical properties including permittivity, mass density, loss tangent when an electric-field interacts with the human-tissue model. Ring-shaped patch with partial-ground and added two-rectangular stubs acting as reflectors is designed for Eight-port and Sixteen-port MIMO antenna offers impedance bandwidth of 34.20GHz-40.0GHz with ground inter-connected and achieves isolation of more than 22.0dB [56]. Also, twelve-port MIMO antenna in 3D sequence with radiating-patch printed in xy, yz and xz plane is designed for two-narrow bands and ultra-wideband applications [57,58]. A 32-port 3D antenna with size of the single element corresponding to $22 \times 20$ mm2 offers super-wideband bandwidth of 3.00–40.0GHz with bandwidth ratio of more than 10:1 [59]. Broader-frequency range antenna generating super-wideband bandwidth of 4.70–67.63GHz achieves impedance bandwidth of 70.4% with peak-gain of 9.36dBi using lumped-mirror terminology [60,61].

A MIMO antenna adhering super-wideband characteristics with inclusive of conformal and SAR analysis is needed to reduce the area on the motherboard designed for RF wireless applications which is not available. Hence motivation to design super-wideband MIMO antenna with flexibility and controllable SAR is designed. In this work, a eight-port MIMO$_{n-s}$ antenna with asymmetric feed fabricated on Rogers-substrate of thickness 0.254mm is analyzed in frequency, time, and far-field region. The dual-band capability achieving WiMAX-band, microwave, and 5G-millimeter wave FR2 bands is used for multi-band applications. Section 2 discusses the evolution of the Oval-shaped single-port antenna Section 3 includes the analysis of the two-port and four-port MIMO$_{n-s}$ antenna and Section 4, the proposed eight-port MIMO$_{n-s}$ antenna.

## 2. Antenna design

In the present fast communication era, the need to transmit information faster with reliability and compact size module (PCB) is the need of the hour. Also, there exist several wireless communication systems that need both transmitter and receiver antenna printed on the same PCB which in turn occupies larger space on the PCB. To reduce the number of antennae for multiple-wireless applications, a single antenna configuration can be designed which can cover several wireless systems in a lower band such as WiMAX (3.30–3.80GHz), and upper-band applications including X-band (8.00GHz-12.0GHz), Ku-band (12.0GHz-18.0GHz), K-band (18.0GHz-27.0GHz), V-band (40.0GHz-75.0GHz) and W-band (75.0GHz-110.0GHz) with inclusive of all the 5G-millimeter FR2 bands. Fig 1a–1g shows an antenna with all the capabilities of covering multiple-wireless applications.

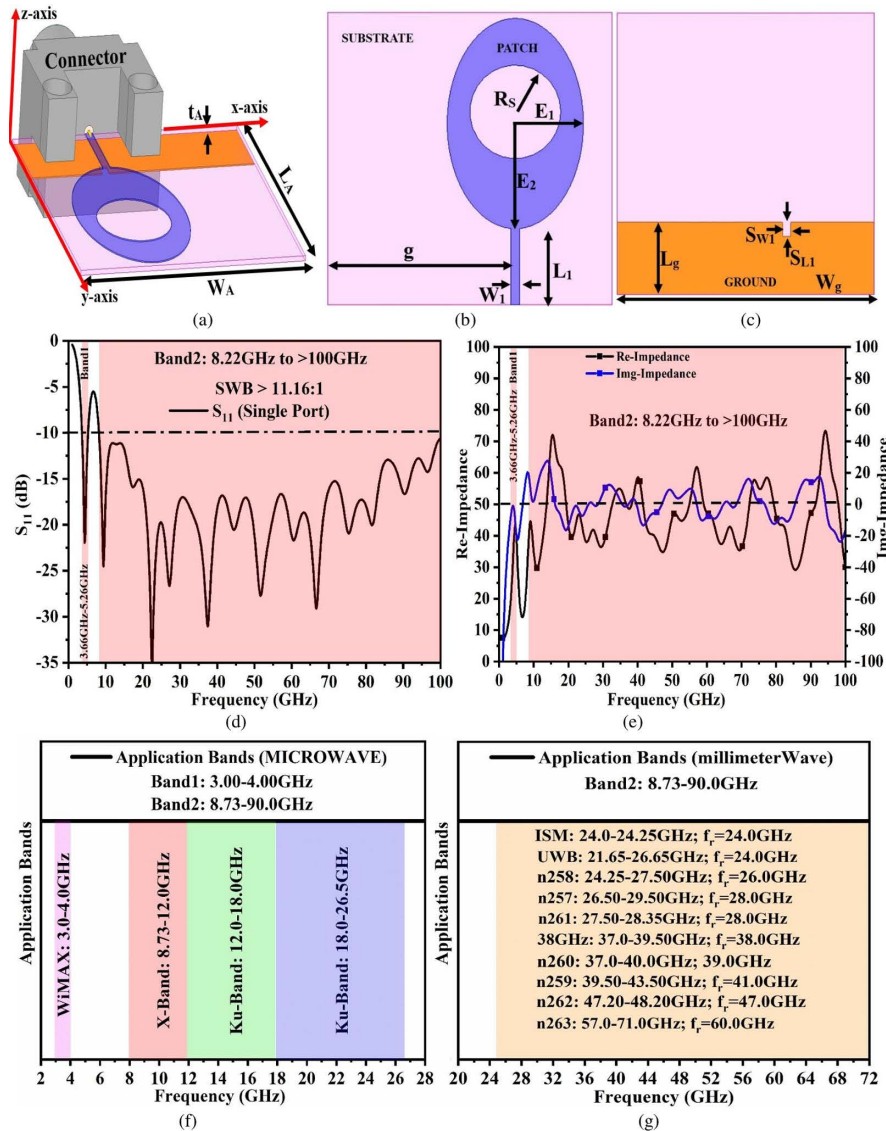

**Fig 1. The proposed dual-band multiband antenna (a) Isometric-details (b)-(c) Front-ground optimal dimensions (d) Simulated S$_{11}$ graph (e) Impedance variation (f)-(g) Application bands.**

## A. Antenna configuration

Fig 1a shows the isometric view of the antenna with an overall dimension of $W_A \times L_A$ mm$^2$. The two planes of the substrate are utilized in printing the radiating surface and the ground plane. The top plane is imprinted by an Oval-patch with an inscribed circular slot and is connected to 50 $\Omega$ matched transmission line. The transmission line is connected to the V-band connector for signal input with a sweep ranging between 1.00GHz-100GHz. Fig 1b shows the front view of the optimal dimension of the elliptical circular-slotted patch with Oval-patch minor and major radius corresponding to $E_1$ and $E_2$ mm. The circular slot of radius $R_s$ mm is etched at the center of the Oval-patch and this forms the radiator of the proposed work. The patch is connected to the transmission line of dimension $W_1 \times L_1$ mm$^2$ and the patch is shifted by a distance of **g** mm from the cornered edge forming asymmetric-feed configuration. The ground which is printed on the opposite surface of the substrate occupies the area of $W_g \times L_g$ mm$^2$ and also consists of the rectangular etched slot of dimension $S_{W1} \times S_{L1}$ mm$^2$ which helps in matching the impedance. The $S_{11}$ graph of the proposed antenna shown in Fig 1a–1c shows the capability of generating two bands namely Band1 and Band2 shown in Fig 1d. The first band, Band1 is very narrow with a bandwidth corresponding to 3.611GHz-5.257GHz. This bandwidth is useful for applications in WiMAX and WLAN bands. The second band, Band2 achieves a bandwidth of 8.128GHz to > 100GHz as observed in Fig 1d. This bandwidth also maintains the super-wideband ratio of more than 10:1 and includes several wireless applications bands both in the microwave and mm-wave band applications. Fig 1e shows the real and imaginary graph with key-frequency points. Fig 1f and 1g list the applications for which the proposed antenna is designed suggesting multi-band characteristics accommodated within the dual-bands. Table 1 lists the optimal antenna parameters which is achieved by EM simulator.

## B. Antenna evolution

The proposed one-port antenna configuration discussed in Sub-Section A achieved dual-bands with wide impedance matching and the wireless bands are useful in several applications listed in Table 1. The final version of the antenna with one-port input, however, is achieved by transforming the Oval patch with the rectangular ground by undergoing iterations as shown in Fig 2a–2e. The first iteration corresponds to the printing of an Oval-patch antenna with minor and major radii corresponding to $E_1 = 4.00$mm and $E_2 = 6.00$mm with rectangular ground printed on opposite plane occupying an area of $6.40 \times 25.0$ mm$^2$ on RogersRTDuroid5880 0.254mm thickness RF-substrate corresponding to Fig 2a. The key factor in achieving the wider-impedance bandwidth is using the partial-ground plane forming monopole configuration. The Oval patch is calculated by the set of equations [42].

$$f_r = \frac{15}{\pi e E_{2eff}} \sqrt{\frac{q}{\varepsilon_r}}$$

(1)

$$E_{2eff} = E_2 \left[ 1 + \left( \frac{2t_A}{\pi \varepsilon_r E_2} \right) p \right]^{\frac{1}{2}}$$

(2)

**Table 1. Antenna design parameters.**

| Par. | Value (mm) | Par. | Value (mm) | Par. | Value (mm) |
|---|---|---|---|---|---|
| $t_A$ | 0.254 | $L_A$ | 25.0 | $W_A = W_g$ | 25.0 |
| $S_{W1}$ | 0.80 | $S_{L1}$ | 0.75 | $E_1$ | 4.00 |
| $E_1$ | 6.00 | $R_s$ | 4.00 | g | 16.10 |
| $W_1$ | 0.80 | $L_1$ | 6.50 | $L_g$ | 6.40 |

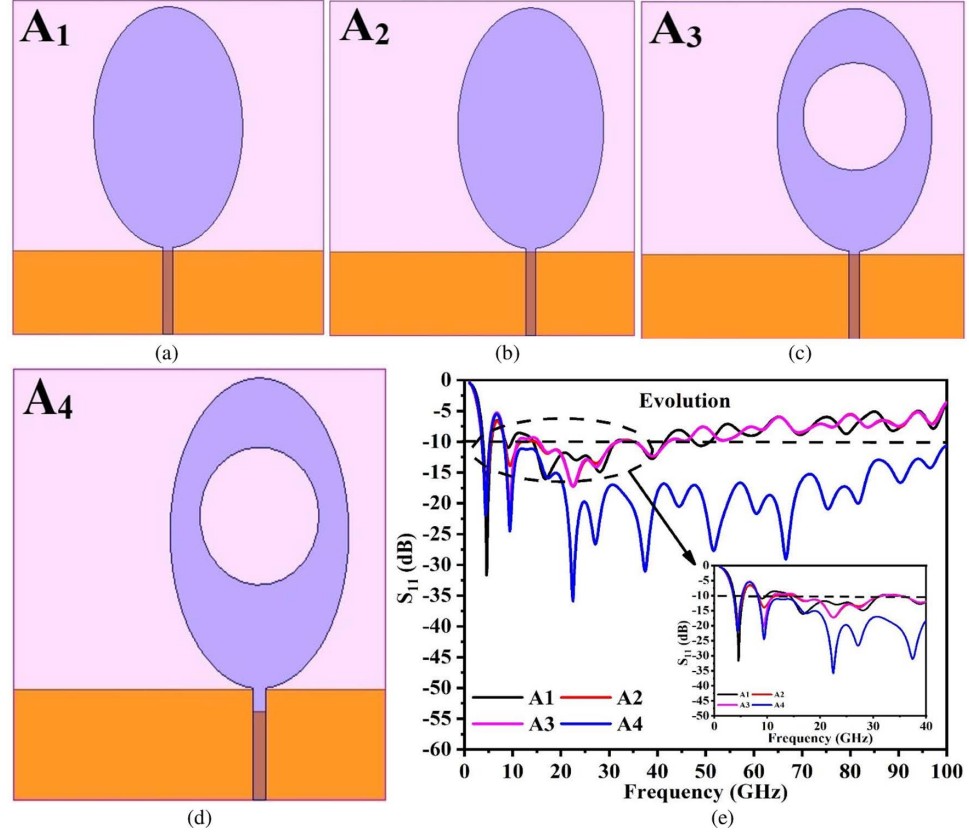

**Fig 2. Single-port-s antenna evolution (a) Antenna A₁ (b) Antenna A₂ (c) Antenna A₃ (d) Antenna A₄ (e) S₁₁ comparison.**

$$p = ln\left(\frac{E_2}{2t_A}\right) + (1.41\varepsilon_r + 1.77)\,\frac{t_A}{E_2}\,(0.268\varepsilon_r + 1.65)$$

(3)

$$q = -0.0049e + 3.7888e^2 - 0.7278e^3 + 2.31e^4$$

(4)

Where $f_r$ is the designed resonance frequency (20.0GHz), q and p are calculated by using the radius of the ellipse, thickness of the substrate ($t_A$), and permittivity of the dielectric material, $E_{2eff}$ is the effective dimension of the ellipse with signal input. Due to the wider-impedance characteristics of the antenna, $f_r = 20.0$GHz is chosen and the corresponding iterations helps in improving the matching of the impedance around the center resonance frequency, $f_r$ GHz. Equation (1)–(4) evaluates the $E_2$ and hence the area of the ellipse required at resonance frequency $f_r = 20.0$GHz. The basic Oval-patch and the partial ground monopole antenna shown in Fig 2a correspond to Antenna A₁ with a matched-impedance bandwidth of 3.80GHz-5.554GHz. The Antenna A₁ shows the capability of achieving a lower band of application as recorded in Table 1. However, to increase and improve the further matching of impedance, the second version shown in Fig 2b designated as Antenna A₂ is achieved by asymmetrically shifting the patch by a distance of 16.10mm from the edge of the substrate which corresponds to bandwidth of 3.60GHz-5.455GHz and 8.30GHz-13.18GHz. The third iteration, Antenna A₃ represented by Fig 2c is achieved by etching a circular slot of radius 4.00mm giving a tri-band. The first bandwidth corresponds to 3.66GHz-5.221GHz, the second bandwidth of 8.30GHz-11.0GHz, and the third bandwidth of

15.55GHz-31.60GHz. The final version of the Antenna $A_4$ is shown in Fig 2d which is achieved by etching the rectangular-slot of dimension $0.80 \times 1.40\,mm^2$ on the rectangular-ground plane. The required bandwidth of 3.50GHz-5.28GHz (Band1) and 8.20GHz-100GHz (Band2) is obtained as shown in Fig 2e which is the objective of the design for multiple-wireless applications.

## C. Time-Domain analysis and parametric study

The time domain study which is achieved by simulation setup becomes very important for the wideband antenna to know the shape of the pulse when transmitted for the different frequency points within the operational bandwidth. The time-domain analysis discusses the key parameters including Fidelity-Factor (FF), impulse response (IR), group delay, isolation, and phase response. The study is carried out by using an identical antenna which is placed at a distance more than $2 \times W_A{}^2/\lambda$ ensuring a far-field region. Fig 3a and 3b shows the arrangement of the identical antenna placed at a distance D = 300mm apart in Face-to-Face (FF) and Side-to-Side (SS) configuration which ensures all the 360° possibility of receiving the signal. The Fidelity-Factor is the measure of the correlation between the transmitted pulse and the received pulse. The Fidelity-Factor is evaluated from the following Equation (5) given below [5]

$$FF = max \left[ \frac{\int_{-\infty}^{+\infty} T_t(t)\,T_r(t+\tau)d\tau}{\sqrt{\int_{-\infty}^{+\infty} |T_t(t)|^2\ dt\ \int_{-\infty}^{+\infty} |T_r(t)|^2\ dt}} \right] \tag{5}$$

where $T_t(t)$ is the transmitted signal and $T_r(t)$ is the received signal. The value of the Fidelity Factor is 1 when the receiver receives a replica of the transmitted pulse with a minimum value of 0.50 being the lower limit. Fig 3c shows the study of the impulse response of the proposed work fed to the transmitter antenna with a normalized amplitude which is a Gaussian pulse. The receiver placed in face-to-face and side-to-side orientations receives the signal with 180° out of phase in comparison with the input phase. The received signal includes a ringing effect. Fig 3d shows the group delay of the super-wideband antenna in both orientations and is defined as a "negative derivative of the phase change concerning frequency". Group delay plays a significant role in understanding the time lag of the input signal at every frequency point. It is also known that when the signal is transmitted by the device, the signal tends to change phase as well as amplitude. The group-delay variation concerning time should be very small for the wide range of frequency bands occupied within the operational bandwidth of a super-wideband antenna, and the variations should not be more than ±1ns indicating the linear-phase is maintained within the operational impedance bandwidth and the linear-phase variations report the distortion in the signal. The group delay is evaluated by Equation (6) given below [5],

$$\tau_g(\omega) = -\frac{d\ \varphi(\omega)}{d\omega} = \frac{d\varphi(\omega)}{2\pi df} \tag{6}$$

where $\varphi$ corresponds to the phase of the transmitted signal, $\omega$ is the frequency (radian/sec). Fig 3e shows the plot of group delay in both orientations (Face-to-Face and Side-to-Side) within the operational bandwidth. The variations are between ±0.50ns indicating the linear phase being constant. Also, there is a slight deviation of group delay between 10.0GHz-15.0GHz which is due to the matching-of impedance starting to get better. Fig 3e observes the isolation between the two orientations placed 300mm apart. In the first band, Band1, the isolation $S_{21}$ (dB) is more than 20.0dB while for Band2, the isolation continues to improve with a value of more than 30.0 dB. Moreover, the isolation or $S_{21}$ (dB) is calculated by Equation (7) given below [5]

$$Isolation_{(port-port1)} = -10 \log_{10} |S_{12}|^2\ (dB) \tag{7}$$

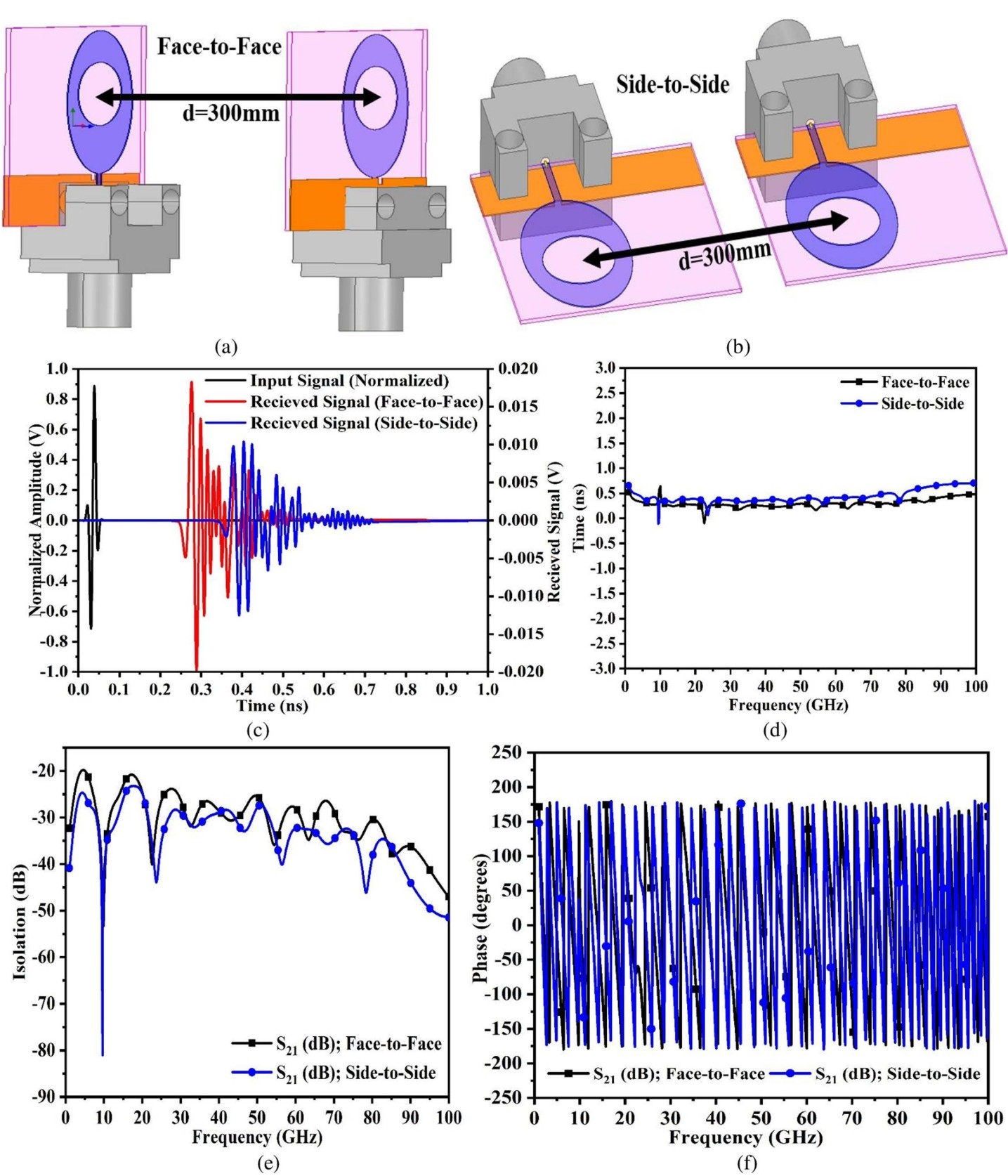

(a)

(b)

(c)

(d)

(e)

(f)

**Fig 3. Time domain analysis (a)-(b) Orientation in Face-to-Face and Side-to-Side configuration (c) Impulse response (d) Group delay (e) Isolation (dB) (f) Phase response.**

Fig 3f shows the phase variations concerning frequency which indicates that there is no phase distortion in the received signal.

In Section 2.1, a super-wideband antenna with a narrow band was discussed. The optimal dimensions for the same are Tabulated in Table 1. However, the optimal values undergo parametric change and finally, the optimal values are obtained for further investigation. Fig 4a and 4b shows the study of key parameters $R_s$ (radius of the etched circle within the Oval patch) shown in Fig 4a and $S_{L1}$ variation plotted in Fig 4b which is the depth of the etched slot in the ground. The first parameter, $R_s$ is changed between two values corresponding to 3.0mm and 5.00mm. The value of $R_s = 3.00$mm extracts impedance bandwidth of 3.411GHz-5.612GHz and 7.92GHz-64.26GHz. However, beyond 64.26GHz, the matching of impedance is very poor and is not useful for any application as the antenna does not meet the -10.0dB matching impedance value. The increase in the area of the circular slot to 4.00mm not only improves the matching of impedance at a lower Band1 but also increases the upper cut-off frequency beyond 100GHz in Band2. Thus, the optimized value of Rs = 4.00mm achieves good impedance-bandwidth in Band1 and Band2. However, for Rs = 5.00mm, the matching of impedance loses its matching and the $S_{11}$ values become poor. The second critical parameter that affects the matching of impedance is the depth of the rectangular etched slot in the ground marked as $S_{L1}$. The value of $S_{L1}$ corresponding to 0.25mm achieves both the narrow-super-wideband but needs more tuning to improve the matching. On the other hand, the higher value of $S_{L1} = 1.25$mm records poor matching beyond 70.0GHz. The value of $S_{L1} = 0.75$mm and $S_{11}$ results offer the required bands in Band1 (3.616GHz-5.26GHz) and Band2 (8.12GHz-100GHz).

## 3. Eight-Port flexible MIMO$_{n-s}$ antenna

In earlier sections, the two-port MIMO$_{n-s}$ antenna and four-port MIMO$_{n-s}$ antenna were discussed and the advantages of multi-port antenna over the one-port antenna were also summarized. This section discusses the proposed eight-port

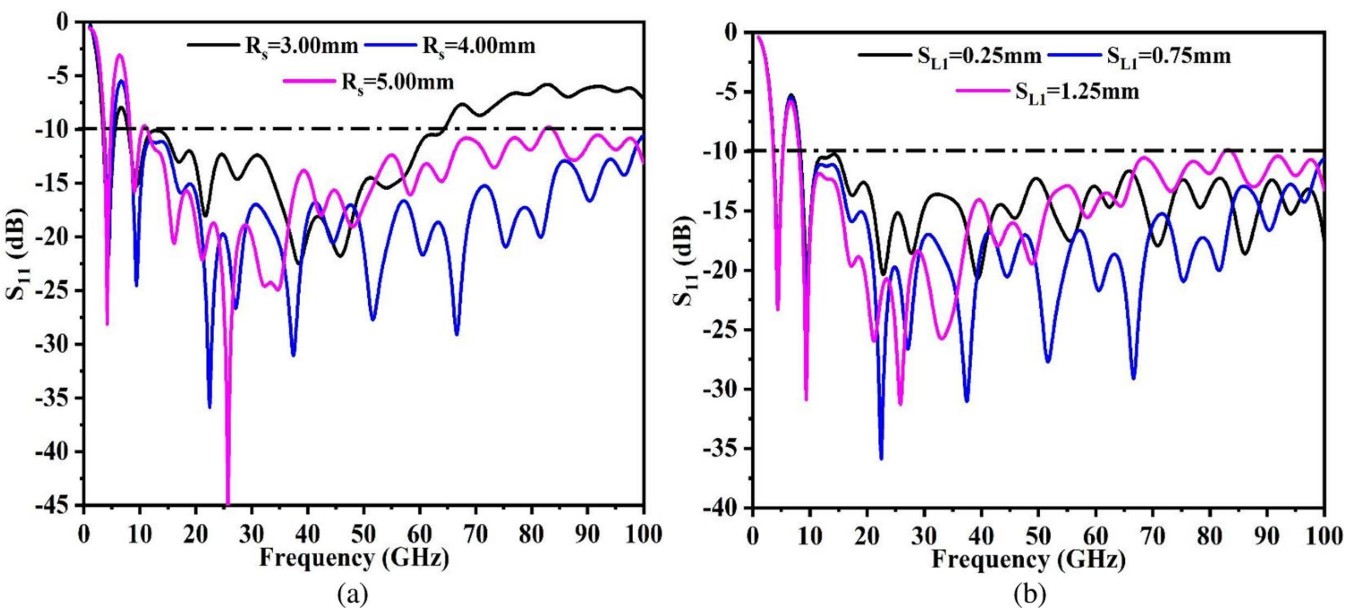

**Fig 4. Parametric study (a) $R_S$ (b) $S_{L1}$.**

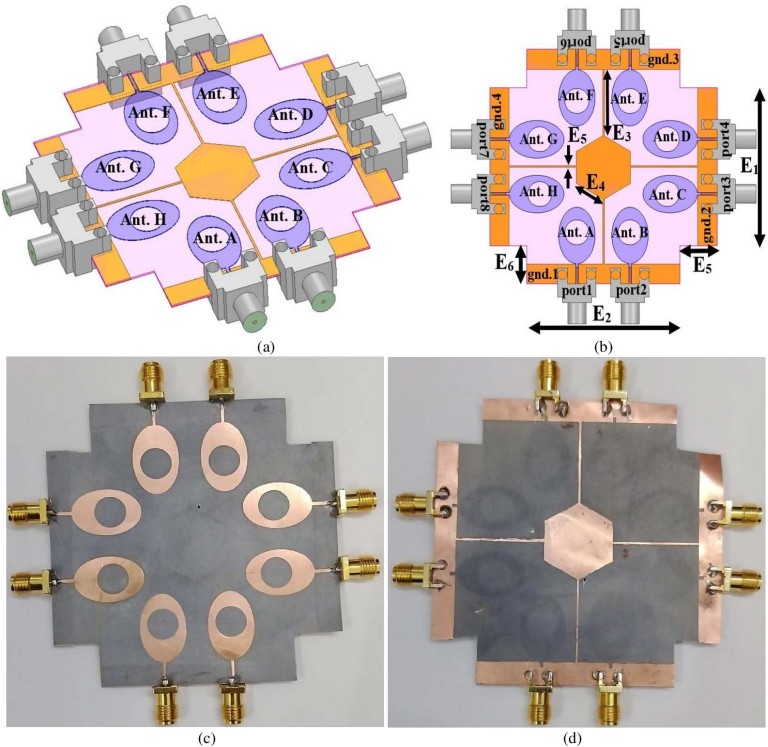

**Fig 5. Eight-Port MIMO$_{n-s}$ Antenna (a)-(b) Eight-port configuration (c)-(d) Prototype photograph.**

MIMO$_{n-s}$ antenna with simulated-measured results comparison, bending capabilities and the ability to be useful for on-body future applications has also been studied and is discussed below.

## A. Eight-Port MIMO$_{n-s}$ antenna

The advantages of MIMO technology over other technologies such as single-input-single-output (SISO) systems were discussed in previous sections. Also, the Shannon theorem as per Equations (8) and (9), the increase in the number of radiating elements not only increases the bandwidth with efficiency, and faster data rate transfer of information but also largely reduces the fading effects suffered by the signal in the fading-environments.

where,

N-No. of antennas at receiver (8 in proposed work)

G-No. of antennas at Transmitter (8 in proposed work)

Ch. N-SWB-Channel Capacity

SNR-Signal-to-Noise Ratio

P$_s$-Signal Power; P$_n$-Noise Power

Fig 5a–5d shows the simulation and the prototype model of the proposed work. Figs 5b and 6a represents the isometric and the detailed optimal dimensions with a front-view transparent snapshot. Fig 5a shows the identical eight Oval-patch with a circular slot connected to the feedline and printed on the top surface. The commonly-connected ground is printed on the opposite surface with a novel modified de-coupling structure. The overall dimension of the antenna is given below,

$$(E_1 + 2(E_6)) \times (E_2 + 2(E_5)) - 4(E_5 \times E_6) = (5625 - 625) = 5000mm^2 \left(70.71 \times 70.71mm^2\right) \tag{10}$$

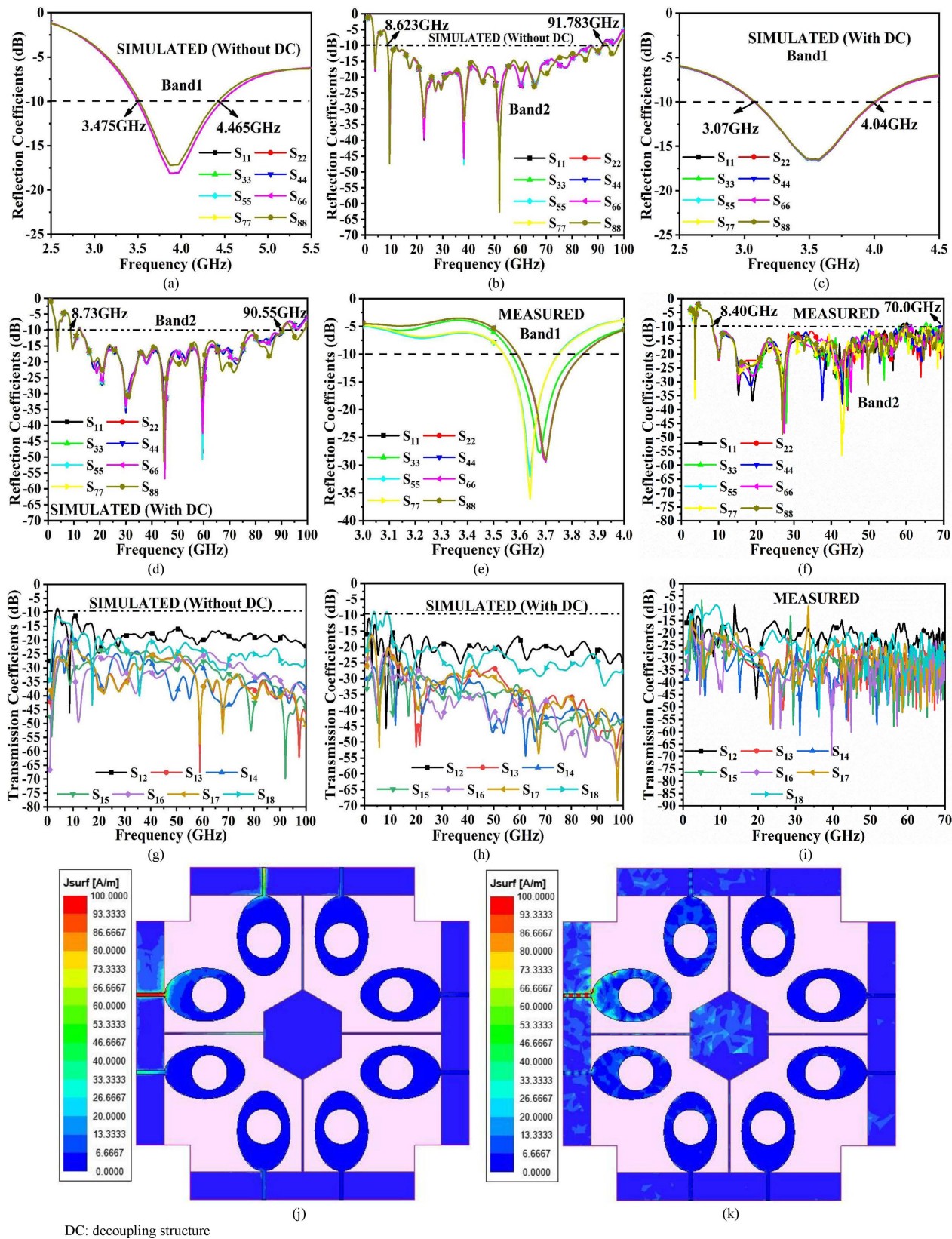

DC: decoupling structure

**Fig 6. Eight-Port MIMO$_{n-s}$ Antenna (a)-(b) Simulated S-parameter (S$_{11}$/S$_{22}$/S$_{33}$/S$_{44}$/S$_{55}$/S$_{66}$/S$_{77}$/S$_{88}$) for Band 1 & Band 2 without Decoupling (c)-(d) Simulated S-parameter (S$_{11}$/S$_{22}$/S$_{33}$/S$_{44}$/S$_{55}$/S$_{66}$/S$_{77}$/S$_{88}$) for Band 1 & Band 2 with Decoupling (e)-(f) Measured S-parameter (S$_{11}$/S$_{22}$/S$_{33}$/S$_{44}$/S$_{55}$/S$_{66}$/ S$_{77}$/S$_{88}$) for Band 1 & Band 2 (g)-(h) Simulated transmission coefficients (S$_{12}$/S$_{13}$/S$_{14}$/S$_{15}$/S$_{16}$/S$_{17}$/S$_{18}$) without and with Decoupling structure (i) Measured transmission coefficients (S$_{12}$/S$_{13}$/S$_{14}$/S$_{15}$/S$_{16}$/S$_{17}$/S$_{18}$); SFD at (j) 3.50GHz (k) 60.0GHz.** DC: decoupling structure.

**Table 2. Optimal dimension values (Fig 6b).**

| Par. | Value (mm) | Par. | Value (mm) |
|---|---|---|---|
| E$_1$ | 50.5 | E$_2$ | 50.5 |
| E$_3$ | 21.25 | E$_4$ | 9.95 |
| E$_5$ | 0.5 | E$_{6=}$E$_7$ | 12.25 |

The dimension values of Equation (10) are tabulated in Table 2.

The Eight-radiating elements are marked as Ant. A, Ant. B, Ant. C, Ant. D, Ant. E, Ant. F, Ant. G, Ant. H. Ant. A and Ant. B are placed adjacent to each other with common-rectangular-ground etched with rectangular slot and is designated as g$_1$. The pair of antennae, Ant. C/Ant. D, Ant. E/Ant. F and Ant. G/Ant. H are placed orthogonal to each other concerning a pair of antenna Ant. A/Ant. B as shown in Fig 5a–5d. The four edges of the substrate are also etched with an area of E$_5$ × E$_6$ = 12.25 × 12.25 mm$^2$. A hexagon patch is placed at the center of the plane with the common-connected ground of side E$_4$ = 9.96mm. All four rectangular grounds (g$_1$, g$_2$, g$_3$, g$_4$) are connected to a hexagon patch via a rectangular strip of dimension E$_3$ × E$_5$ = 0.5 × 20.74 mm$^2$. Fig 5c and 5d shows the fabricated prototype with a front view inclusive of the radiating patch and the common-connected ground with connectors connected to the microstrip feedline for input-signal transmission.

Fig 6a–6k shows the detailing of the Eight-port MIMO$_{n-s}$ antenna with a simulation environment and the fabricated prototype with S-parameters.

Fig 6a and 6b records the simulation of reflection coefficient values without de-coupling (DC) structure and also marks the Band1 and Band2 with a sweep ranging between 2.50GHz-5.50GHz and 1GHz-100GHz. The graph, Fig 6a and 6b records the bandwidth between 3.475GHz-4.465GHz (Without DC) and 3.07GHz-4.04GHz (With DC) in Band1. The absence of the decoupling structure records the shifting of the required WiMAX band as shown in Fig 7a. However, the Band2 records bandwidth of 8.623GHz-91.783GHz (Without DC) and 8.73GHz-90.55GHz (With DC) respectively as noted in Fig 6b and 6e. In Fig 6e and 6f records the sweep in the frequency range is between 1.0GHz-70.0GHz due to the limitations of the available measurement facilities. The Band1 measured bandwidth corresponds to 3.59GHz-3.85GHz and Band2 of bandwidth of 8.40GHz-70.0GHz respectively. Hence, the addition of a de-coupling structure serves two purposes, one improves the isolation between the inter-spaced radiating elements and the other solves the problem of common ground [62]. Fig 6g and 6h shows the simulation for both, Without DC and With DC including S-parameters S$_{12}$, S$_{13}$, S$_{14}$, S$_{15}$, S$_{16}$, S$_{17,}$ and S$_{18}$ representing isolation between port1-port2, port1-port3, port1-port4, port1-port5, port1-port6, port1-port7, and port1-port8. The isolation improves with an additional decoupling (DC) structure in comparison to the isolation without decoupling (DC) structure. The comparison of Fig 6g and 6h indicates that the isolation between 10GHz-60GHz is better than the MIMO antenna without de-coupling structure. Moreover, the isolation between port1-port2 in simulated and measured environments is more than 15.0dB while the isolation for remaining ports is better than the former port1-port2 (S$_{12}$). Fig 6i represents the measured transmission coefficient inclusive of the de-coupling structure which is more than 15.0dB in Band1 and Band2 respectively. Fig 6j and 6k shows the distribution of surface current with excited port1 for frequencies of 3.50GHz and 60.0GHz. As per the observations, the radiations radiated by Ant. A has no impact on the neighboring radiating antenna which is due to the de-coupling structure providing the additional path for the flow of current and the orthogonal arrangement of the pair of antennas as shown in Fig 5a–5d.

## B. Eight-Port MIMO$_{n-s}$ antenna diversity-performance (Simulated-measured)

The MIMO$_{n-s}$ Eight-port configuration are shown in Fig 6a–6o explained the arrangement of the radiating elements and corresponding results related to S-parameters and SFD. To better understand the interaction between each of the radiating elements, the simulated-measured diversity-performance parameters including Envelope-Correlation-Coefficient$_{narrow-super\ wideband}$ (ECC$_{n-s}$), Diversity-Gain$_{n-s}$ (DG$_{n-s}$), are investigated and plotted.

Total-Active-Reflection-Coefficientn-s (TARC$_{n-s}$) and Channel-Capacity-Lossn-s (CCL$_{n-s}$) as shown in Fig 7a–7v. In all the parameter calculations of ECCn-s, DGn-s, TARCn-s, and CCL$_{n-s}$ are calculated between port1-port2, port1-port3, poer1-port4, port1-port5, port1-port6, port1-port7 and port1-port8 respectively. The correlation between the adjacent MIMO$_{n-s}$ antenna is studied by calculating ECC$_{n-s}$ between any two-port and is given by [34,36].

$$ECC_{n-s} = \rho_e(m, s, N) = \frac{\left| \sum_{n=1}^{N} S_{m,n}^* \, S_{n,s} \right|^2}{\pi_{k=(m,s)} \left[ 1 - \sum_{n=1}^{N} S_{m,n}^* \, S_{n,k} \right]} \tag{11}$$

Equation 11 shows the calculation of ECC$_{n-s}$ between the $m^{th}$ and the $s^{th}$ port and in general is calculated by the following equation.

$$ECC_{n-s} = \frac{\left| S_{mm}^* S_{ms} + S_{sm}^* S_{ss} \right|^2}{\left(1 - |S_{ii}|^2 - |S_{sm}|^2\right)\left(1 - |S_{ss}|^2 - |S_{ms}|^2\right)} \tag{12}$$

For two-port and four-port$_{n-s}$, ECC$_{n-s}$ is given by,

$$ECC_{n-s(Two\ Port)} = \frac{\left| S_{11}^* S_{12} + S_{12}^* S_{22} \right|^2}{\left(1 - |S_{11}|^2 - |S_{21}|^2\right)\left(1 - |S_{12}|^2 - |S_{22}|^2\right)} \tag{13}$$

$$ECC_{n-s(Four\ Port)} = \frac{\left| S_{11}^* S_{12} + S_{12}^* S_{22} + S_{13}^* S_{32} + S_{14}^* S_{42} \right|^2}{\left(1 - |S_{11}|^2 - |S_{21}|^2 - |S_{31}|^2 - |S_{41}|^2\right)\left(1 - |S_{12}|^2 - |S_{22}|^2 - |S_{32}|^2 - |S_{42}|^2\right)} \tag{14a}$$

$$ECC_{n-s} = \rho_{e(n-s)} = \frac{\left| \iint_{4\pi} \left[ \overrightarrow{F_i}(\theta, \phi) * \overrightarrow{F_j}(\theta, \phi) \right] d\Omega \right|}{\iint_{4\pi} \left| \overrightarrow{F_i}(\theta, \phi) \right|^2 d\Omega \iint_{4\pi} \left| \overrightarrow{F_j}(\theta, \phi) \right|^2 d\Omega} \tag{14b}$$

$$ECC_{n-s} = \frac{\left| \int_0^{2\pi} \iint_0^{\pi} E_1(\theta, \phi).E_2^*(\theta, \phi) Sin(\theta) d\theta d\phi \right|^2}{\left( \int_0^{2\pi} \int_0^{\pi} |E_1(\theta, \phi)|^2 Sin(\theta) d\theta d\phi \right)\left( \int_0^{2\pi} \int_0^{\pi} |E_2(\theta, \phi)|^2 Sin(\theta) d\theta d\phi \right)} \tag{14c}$$

The simulated and measured ECC$_{n-s}$ are calculated between Port 1-Port 2 (ECC$_{(m-s)12}$), Port 1-Port 3 (ECC$_{(n-s)13}$), Port 1-Port 4 (ECC$_{(n-s)14}$), Port 1-Port 5 (ECC$_{(n-s)15}$), Port 1- Port 6 (ECC$_{(n-s)16}$), Port 1-Port 7 (ECC$_{(n-s)17}$) and Port 1-Port 8 (ECC$_{(n-s)18}$). The ECC$_{n-s}$ can also be calculated by using a radiation pattern and using Equations 14(b) and 14(c). Fig 7a

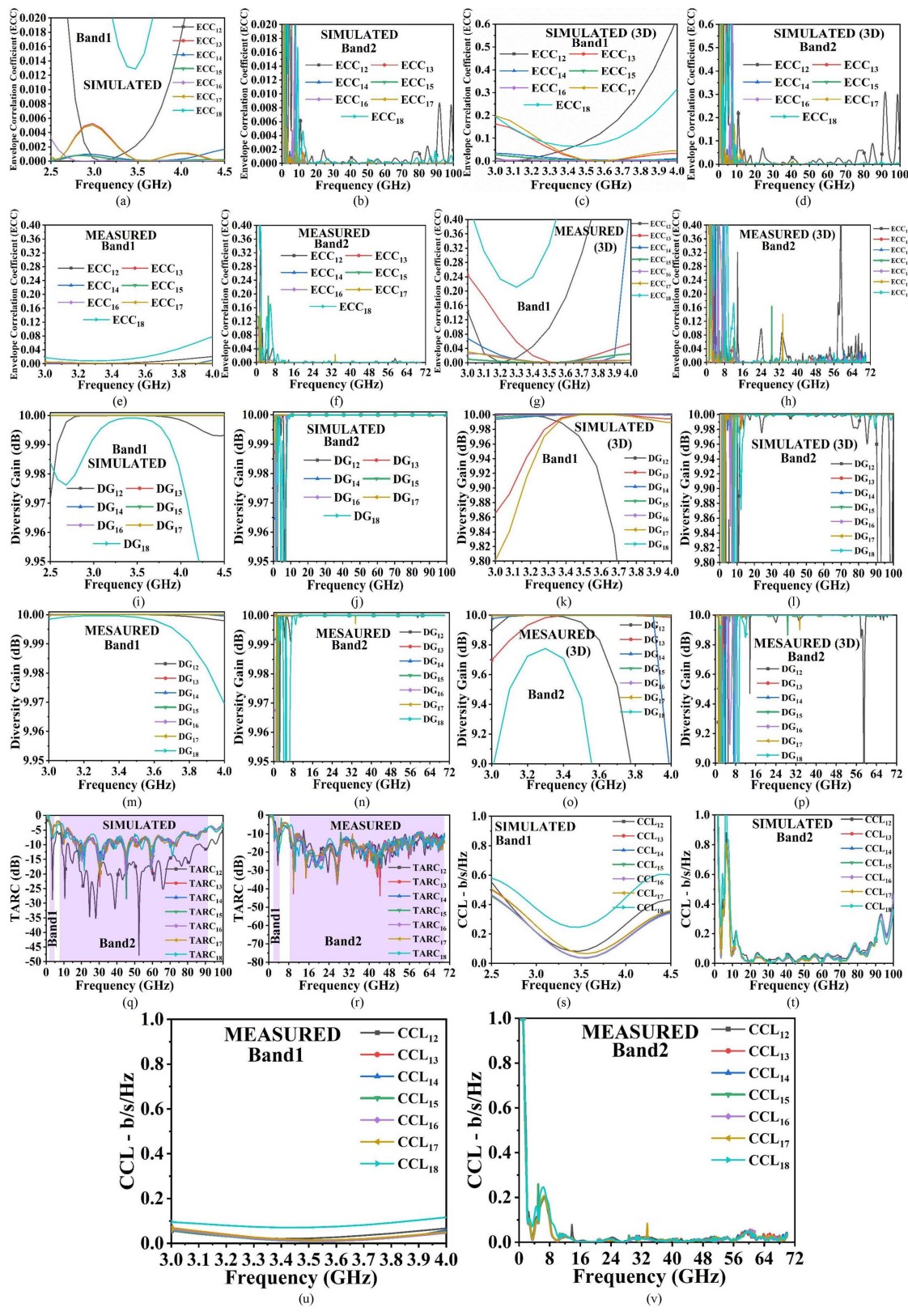

**Fig 7. Diversity-performance (Eight-Port MIMO$_{n-s}$ Antenna) (a)-(b) Simulated ECC$_{n-s}$ (Band 1 & Band 2)-S Parameter (c)-(d) Simulated ECC$_{n-s}$ (Band 1 & Band 2)-3D Radiation Pattern (e)-(f) Measured ECC$_{n-s}$ (Band 1 & Band 2)-S Parameter (g)-(h) Measured ECC$_{n-s}$ (Band 1 & Band 2)-3D Radiation Pattern (i)-(j) Simulated DG$_{n-s}$ (Band 1 & Band 2)-S Parameter (k)-(l) Simulated DG$_{n-s}$ (Band 1 & Band 2)-3D Radiation Pattern (m)-(n) Measured DG$_{n-s}$ (Band 1 & Band 2)-S Parameter (o)-(p) Measured DG$_{n-s}$ (Band 1 & Band 2)-3D Radiation Pattern (q)-(r) Simulated & Measured TARC$_{n-s}$ (s)-(t) Simulated CCL$_{n-s}$ (Band 1 & Band 2) (u)-(v) Measured CCL$_{n-s}$ (Band 1 & Band 2).**

and 7b and Fig 7c and 7d calculate the Simulated ECC$_{n-s}$ by S-parameter and 2D radiation method while the measured ECC$_{n-s}$ by both the methods are shown in Fig 7e and 7f and Fig 7g and 7h.

The Diversity-Gain$_{n-s}$ (DG$_{n-s}$) is calculated to quantify the performance characteristics of the diversity scheme used (spatial, polarization, or radiation diversity) and also signifies the quality & reliability of the MIMO elements. The DG$_{n-s}$ is related to ECC$_{n-s}$ by the following formula [34].

$$DG_{n-s} = 10\sqrt{1 - \left|\rho_{e(n-s)}\right|^2}$$

(15)

The standardized values for Diversity-Gain$_{n-s}$ are > 9.950 dB. The simulated and measured DGn-s in both methods are plotted in Fig 7i–7p. The ratio involved between the total reflected power (TRP) irrespective of the radiating elements and the total power impinged on the patch is known as TARC$_{n-s}$ which are calculated from Equations (16)–(20) [34].

$$\Gamma_{a(n-s)}^{t} = \frac{Available\ Power\ (AP) - Radiated\ Power\ (RP)}{Available\ Power\ (AP)}$$

(16)

In general, for the ideal-case (loss-less) MIMO$_{n-s}$ system, the values for TARC$_{n-s}$ = 0.0dB are calculated for any combination of two ports [34].

$$TARC_{n-s} = \frac{\sqrt{\sum_{i=1}^{N}\left|m_i\right|^2}}{\sqrt{\sum_{i=1}^{N}\left|s_i\right|^2}}$$

(17)

[m] and [n] correspond to incident and reflected power in dB. The TARC$_{n-s}$ concerning phase values of S-parameters are given below

$$b_1 = S_{11}a_1 + S_{12}a_2 = S_{11}a_0e^{j\theta_1} + S_{12}a_0e^{j\theta_2} = a_1\left(S_{11} + S_{12}e^{j\theta}\right)$$

(18)

$$b_2 = S_{21}a_1 + S_{22}a_2 = S_{21}a_0e^{j\theta_1} + S_{22}a_0e^{j\theta_2} = a_1\left(S_{21} + S_{22}e^{j\theta}\right)$$

(19)

Combining Equations, TARC$_{n-s}$

$$TARC_{n-s} = \frac{\sqrt{\left|S_{11} + S_{12}e^{j\theta}\right|^2 + \left|S_{21} + S_{22}e^{j\theta}\right|^2}}{\sqrt{2}}$$

(20)

Assuming phase for both incident and reflected wave to be θ=0°, the TARC$_{n-s}$ are calculated for Port1-Port2, Port1-Port3, Port1-Port4, Port1-Port5, Port1-Port6, Port1-Port7 and Port1-Port8 which corresponds to more than 8.0dB and 15.0dB for simulated and measured values represented by Fig 7q and 7r. The signal transmitted between Tx to Rx does suffer some loss in the MIMO$_{n-s}$ system when the proposed antenna is deployed in the wireless communication

**Table 3. Optimal dimension values.**

| Diversity Parameters | Simulated | | | | Measured | | | | Ideal Values |
|---|---|---|---|---|---|---|---|---|---|
| | S Parameter (dB) | | 3D Radiation Pattern | | S Parameter | | 3D Radiation Pattern | | |
| | Band1 | Band2 | Band1 | Band2 | Band1 | Band2 | Band1 | Band2 | |
| ECC | <0.05 | <0.001 | <0.38 | <0.15 | <0.10 | <0.02 | <0.30 | <0.40 | <0.50 |
| DG (dB) | >9.995 | >9.998 | 9.98 | 9.96 | >9.9995 | >9.9998 | 9.995 | 9.58 | >9.95 |
| TARC (dB) | <-8.0 | <-15.0 | NA | NA | <-8.0 | <-15.0 | NA | NA | <0.0 |
| CCL (b/s/Hz) | <0.30 | <0.015 | NA | NA | <0.10 | <0.001 | NA | NA | <0.40 |

environment and this is signified by the loss of bits in the channel shown in Fig 7s–7v which should be less than 0.40bits/second/Hz. This is evaluated by [34],

$$CCL_{n-s} = -\log_2 \det\left(\alpha^s\right) \tag{21}$$

where

$$\rho_{mm(n-s)} = 1 - \sum_{n=1}^{4} \left|S_{mn}\right|^2 \tag{22}$$

$$\rho_{ms(n-s)} = -\left(S_{mm}^* S_{ms} + S_{sm}^* S_{ms}\right) \tag{23}$$

Table 3 shows simulated and measured diversity parameters. The ECC$_{n-s}$ and DG$_{n-s}$ are calculated by S-parameter and 3D radiation pattern method. The 3D radiation pattern method includes losses in comparison to the S-parameter and the values are poorer in comparison with the S-parameter method but are below the ideal values. The TARC$_{n-s}$ and CCL$_{n-s}$ are also below 0.0dB and 0.40b/s/Hz for both, simulated and measured frequency ranges in Band1 and Band2 respectively.

## C. Bending-analysis

The emergence of flexible electronics has opened the gate for antennae in the modern era, with the applications extending to devices to multiple-non-flat surfaces, which also include the shape of the body. Human health is possible to be monitored by using flexible-antenna for on-body applications which shows the ability to detect microstructures-deformations. The fourth (4G) and the fifth (5G) generation have been able to encourage higher data rate transfer. The wearable antenna with flexible configuration is helpful in applications such as medical and and entertainment. Medical diagnosis and treatment are carried out by using the device with an embedded flexible antenna, while in entertainment, this antenna can be used for gaming and virtual-reality applications. The proposed eight-port MIMO$_{n-s}$ antenna uses Rogers RTDuroid5880 substrate with a thickness of 0.254mm which can be easily used for flexible applications but needs to ensure that the bending does not compromise the operational bandwidth. Fig 8a–8d give insight into the bending characteristics of the antenna at different angles corresponding to 0°, 15°, 30° and 45° with corresponding -10.0dB bandwidth plotted in Fig 8e. Due to the symmetrical structure, the antenna can be bent either on the x-axis or y-axis and in this work, the x-axis bending of the antenna is considered. Fig 8e concludes that at all four bending angles, the MIMO$_{n-s}$ antenna maintains the operational bandwidth with Band1 = 3.12–4.02GHz and Band 2 = 8.728–90.30GHz.

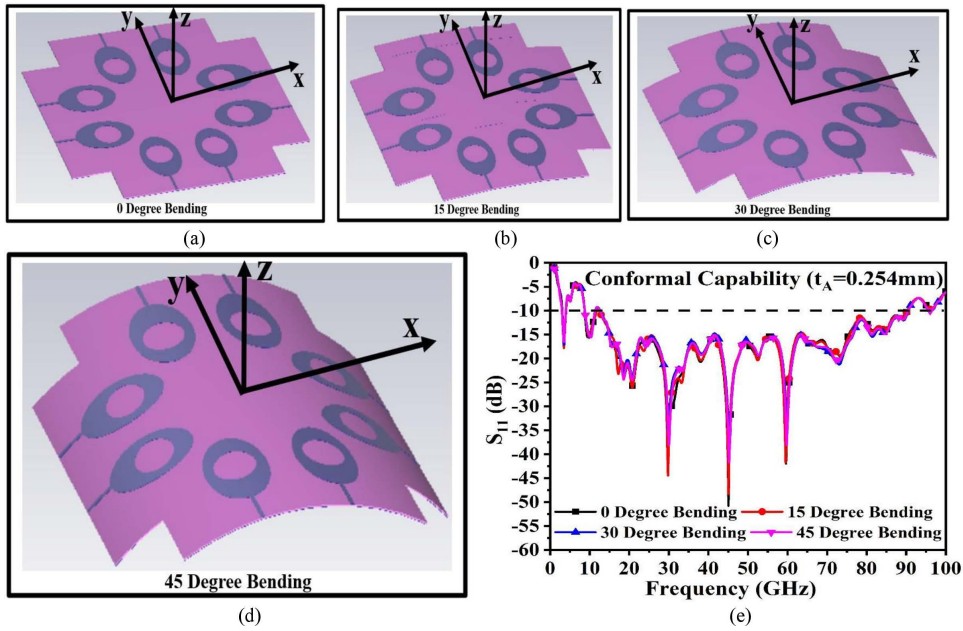

**Fig 8. Bending analysis of 8-Port MIMO$_{n-s}$ antenna at (a) 0° (b) 15° (c) 30° (d) 45°.**

## D. SAR analysis

A parameter, Specific-Absorption-Rate needs to be calculated by designing the human-tissue model with known electrical properties and placing the antenna above the tissue. Fig 9a–9m illustrates the modeling of the tissue and the MIMO$_{n-s}$ eight-port antenna placed over it. Fig 9a shows that the tissue model consists of three layers namely skin, fat, and muscle. The antenna to be tested for SAR analysis is placed with a gap of 5.00mm from the surface of the skin. The skin, fat, and muscle thickness correspond to 2.00mm, 5.00mm, and 5.0mm respectively as depicted in Fig 9a and 9b. Table 4 tabulates the electrical properties of the tissue model: skin, fat, and muscle whose values vary with frequency. However, the mass density (Kg/m³) for skin = 1109 Kg/m³, fat = 911 Kg/m³, and muscle = 1090 Kg/m³ remains constant irrespective of change in frequency. The SAR is calculated by the Equation (24) given below [20].

$$SAR_{n-s} = \frac{\sigma |E|^2}{\rho}$$

(24)

where

σ-conductivity of body tissue (S/m)

E-applied electric field (V/m)

P-mass density of the body tissue (Kg/m³)

Fig 9c–9m shows the simulation of the tissue model at the frequency values corresponding to 3.50GHz, 10.0GHz, 15.0GHz, 24.0GHz, 26.0GHz, 28.0GHz, 38.0GHz, 39.0GHz, 41.0GHz, 47.0GHz and 60.0GHz. Power of 1mW is applied in simulation with an average of 1g of the tissue. The SAR calculated at different frequency points over the tissue model are tabulated in Table 5 with SAR values not more than 1.60W/Kg. Table 5 also shows that the frequency points chosen correspond to the center frequency of the application bands in WiMAX, X, Ku, and FR2 millimeter-wave bands. This suggests that the proposed 8-port MIMO$_{n-s}$ antenna is well-suitable for on-body applications for existing and future mmWave applications.

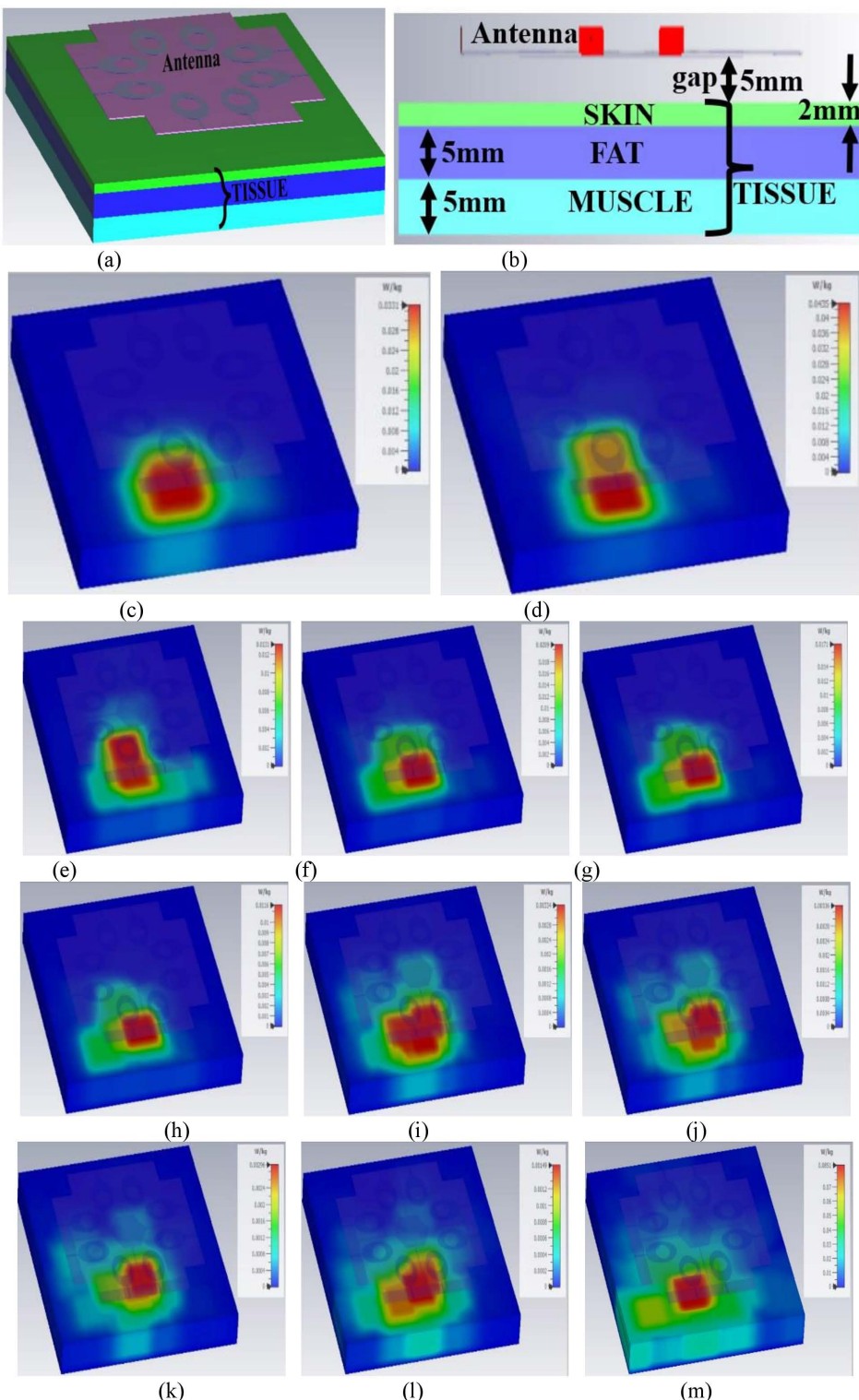

**Fig 9. SAR<sub>n-s</sub> Analysis of 8-Port MIMOn-s antenna at (a) The Tissue model (b) Dimensions of the tissue model; SARn-s calculations at (c) 3.50GHz (d) 10.0GHz (e) 15.0GHz (f) 24.0GHz (g) 26.0GHz (h) 28.0GHz (i) 38.0GHz (j) 39.0GHz (k) 41.0GHz (l) 47.0GHz (m) 60.0GHz.**

**Table 4. Electrical properties of tissue at different frequency [54,55].**

| Tissue | Frequency (GHz) | Electrical permittivity | Conductivity | Loss Tangent | Density (Kg/m³) |
|---|---|---|---|---|---|
| **Skin** | 3.50 | 37.0 | 2.03 | 0.2813 | 1109 |
| | 10.0 | 31.3 | 8.02 | 0.46038 | |
| | 15.0 | 26.4 | 13.9 | 0.62855 | |
| | 20.0 | 22.0 | 19.2 | 0.78675 | |
| | 24.0 | 19.0 | 22.9 | 0.90073 | |
| | 26.0 | 17.7 | 24.4 | 0.8646 | |
| | 28.0 | 16.5 | 25.8 | 1.0016 | |
| | 38.0 | 12.3 | 31.0 | 1.0291 | |
| | 39.0 | 12.0 | 31.4 | 1.2089 | |
| | 41.0 | 11.4 | 32.2 | 1.236 | |
| | 47.0 | 9.97 | 33.9 | 1.3003 | |
| | 60.0 | 7.98 | 36.4 | 1.3673 | |
| **Fat** | 3.50 | 10.5 | 0.422 | 0.154 | 911 |
| | 10.0 | 8.80 | 17.1 | 0.22857 | |
| | 15.0 | 7.79 | 2.76 | 0.26309 | |
| | 20.0 | 7.01 | 3.72 | 0.28248 | |
| | 24.0 | 6.50 | 4.43 | 0.29076 | |
| | 26.0 | 6.29 | 4.74 | 0.29319 | |
| | 28.0 | 6.09 | 5.06 | 0.29471 | |
| | 38.0 | 5.33 | 6.36 | 0.29331 | |
| | 39.0 | 5.27 | 6.47 | 0.29262 | |
| | 41.0 | 5.16 | 6.69 | 0.29103 | |
| | 47.0 | 4.86 | 7.30 | 0.2851 | |
| | 60.0 | 4.40 | 8.39 | 0.26925 | |
| **Muscle** | 3.50 | 51.4 | 2.56 | 0.25533 | 1090 |
| | 10.0 | 42.8 | 10.6 | 0.44666 | |
| | 15.0 | 36.4 | 17.9 | 0.5904 | |
| | 20.0 | 31.0 | 24.7 | 0.71649 | |
| | 24.0 | 27.3 | 29.5 | 0.80483 | |
| | 26.0 | 25.9 | 31.6 | 0.84508 | |
| | 28.0 | 24.4 | 33.6 | 0.88284 | |
| | 38.0 | 19.1 | 41.8 | 1.0382 | |
| | 39.0 | 18.6 | 42.5 | 1.051 | |
| | 41.0 | 17.9 | 43.8 | 1.0752 | |
| | 47.0 | 15.9 | 47.2 | 1.1378 | |
| | 60.0 | 12.9 | 52.8 | 1.231 | |

## E. Far-field analysis and comparison of the proposed work with existing literature

Fig 10a–10l shows the analysis of the eight-port MIMO$_{n-s}$ antenna in the far-field region. The two results, peak-realized gain and 2-D radiation patterns in principal planes are plotted. Fig 10a shows the prototype placed within the anechoic chamber with the horn antenna as the transmitter and the fabricated prototype as a receiver. Fig 10b calculates the peak-realized gain in the simulation environment varies between 2.50dBi-16.35dBi with an average peak gain of 12.90dBi. The measured peak-realized gain is calculated at selected frequency values with

**Table 5. SARn-s values.**

| Sl. No. | Frequency (GHz) | Application Band | SAR (W/Kg) | Ideal Values |
|---------|-----------------|------------------|------------|--------------|
| 1 | 3.50 | WiMAX | 0.0331 | 1.60 W/Kg |
| 2 | 10.0 | X-Band | 0.0435 | |
| 3 | 15.0 | Ku-band | 0.0131 | |
| 4 | 24.0 | ISM band | 0.0209 | |
| 5 | 26.0 | n258 | 0.0171 | |
| 6 | 28.0 | n257/n261 | 0.0116 | |
| 7 | 38.0 | n260 | 0.00334 | |
| 8 | 39.0 | n260 | 0.00336 | |
| 9 | 41.0 | n259 | 0.00296 | |
| 10 | 47.0 | n262 | 0.00149 | |
| 11 | 60.0 | n263 | 0.0851 | |

peak-gain of 2.965dBi at 3.50GHz, 3.556dBi at 10.0GHz, 6.3256dBi at 15.0GHz, 8.5698dBi at 20.0GHz, 10.1258dBi at 24.0GHz, 12.6987dBi at 28.0GHz, 11.5895dBi at 38.0GHz, 13.2658dBi at 39.0GHz, 14.56258dBi at 41.0GHz, 15.2658dBi at 43.0GHz, 14.85635dBi at 47.0GHz and 16.1258dBi at 60.0GHz. The average measured gain corresponds to 10.18483dBi in both bands. Fig 10c–10l shows the simulated-measured 2-D radiation pattern in principal planes in the E-H plane at 3.50GHz, 10.0GHz, 28.0GHz, 38.0GHz, and 60.0GHz respectively. The 2-D radiation pattern corresponds to bi-directional for E-filed or $\phi=0°$ and omnidirectional in H-field or $\phi=90°$ at lower frequencies corresponding to 3.50GHz and 10.0GHz. At higher frequency values, the higher-order modes beyond 10.0GHz become predominant and add to the distortion of the shape in the radiation patterns as Fig 10g–10l. The Eight-port MIMO$_{n-s}$ antenna which is the objective of the work is achieved by single-port configuration. The proposed work radiates in in narrow bandwidth and the in super-wideband bandwidth. Table 5 shows the comparative study with existing literature. Firstly, the proposed size of the antenna generates the narrow as well as super-wideband bandwidths which is not seen with the other Eight-port antenna shown in Table 6. Also, the conformal analysis and SAR calculation are absent in the eight-port configuration while the proposed work includes both the analysis at several frequency values. In addition, the other Eight-port MIMO antenna also doesn't include bending analysis. The features such as dual-band and a wider range of applications outclass the other previously Eight-port MIMO antenna configurations compared in Table 6.

## 4. Conclusions

An eight-port MIMO$_{n-s}$ antenna with narrow-super wideband capability is investigated for multiple-band applications which includes Oval-defected patch and defected-ground-structure with asymmetric-feed. The two bands are achieved for WiMAX applications and the other for X-, Ku, K, and 5G-FR2 millimeter wave bands. The bending analysis is also studied with preserved -10.0dB bandwidth when bent at 15, 30, and 45. The SAR analysis is also done at key frequency values with controllable SAR below 1.60W/Kg for 1g of the tissue and 50 mW of input power. The diversity parameters The diversity performance is also evaluated with measured ECCn-s < 0.02, DGn-s > 9.9998dB, TARCn-s < -8.0dB, and CCLn-s < 0.10b/s/Hz are below the ideal standard values and due to the multiple-characteristics, the proposed work is well suited for wireless on-body applications for a present and future scenario.

This article does not contain any studies with human participants or animal performed by any of the authors. Informed consent No human or animals were involved. All applicable international, national, and/ or institutional guidelines for the care and use of animals were followed.

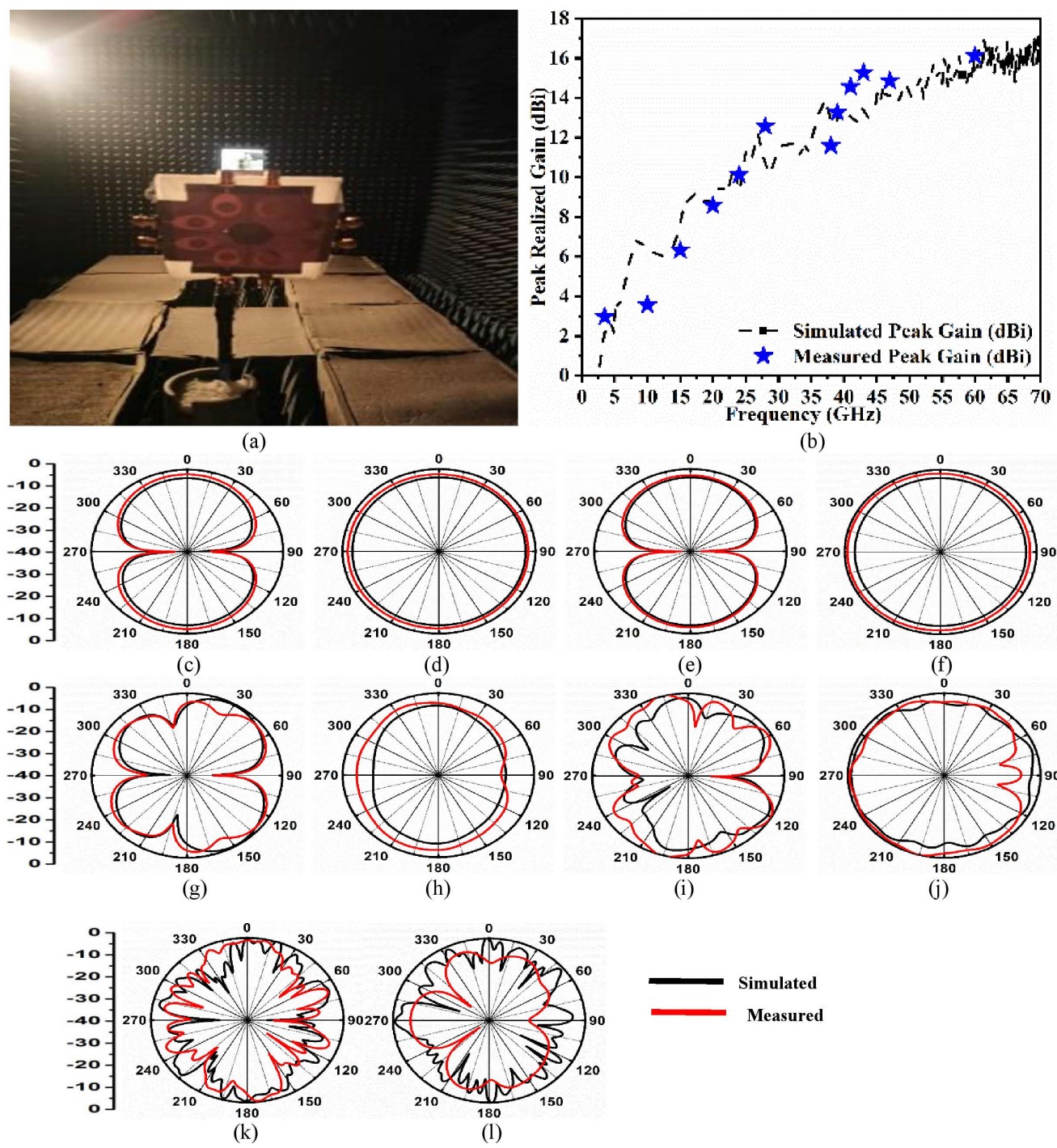

**Fig 10. FAR-Field result discussion (a) Prototype placed within the Anechoic Chamber (b) Simulated-Measured gain comparison; 2-D Radiation pattern in E-H plane at (c)-(d) 3.50GHz (e)-(f) 10.0GHz (g)-(h) 28.0GHz (i)-(j) 38.0GHz (k)-(l) 60.0GHz.**

**Table 6. Proposed MIMOn-s work comparison with existing literature.**

| Ref. | Size of the Antenna (mm²) | Substrate/ Thickness (mm) | Bandwidth (GHz) | No. of Ports | Isolation | ECC | DG (dB) | TARC (dB) | CCL (b/s/Hz) | Bending Capability | SAR Analysis (W/Kg) |
|---|---|---|---|---|---|---|---|---|---|---|---|
| [23] | 68×80 6.80λo×5.70λo | RO4350TM 0.80 | 11.1-14.89 24.23-25.53 26.81-27.12 30.50-33.75 | 02 | 15.0 | 48×10⁻⁴ | 9.98 | NC | 23×10⁻⁵ | NO | 0.2230 at 25.50GHz |
| [28] | 56×56 0.60λo×0.60λo | LCP 0.10 | 2.54-3.56 4.28-4.97 5.37-8.85 | 04 | 20.0 | 0.13 | 9.95 | 10.0 | NC | YES 30mm 45mm | 1.94 at 3.5GHz 0.78 at 4.7GHz 0.46 at 6.5GHz |
| [32] | 40×40 0.54λo×0.54λo | Jeans Textile Substrate 1.50 | 3.18-4.60 8.89-12.0 | 04 | 20.0 | 0.12 | NC | 6.0 | 0.19 | YES 28mm 29mm 30mm 31mm 32mm | 1.0 at 10GHz |
| [34] | 20×20 0.70λo×0.70λo | Rogers5880 0.254 | 8.31-36.14 | 04 | 10.0 | 0.40 | 9.999 | -4.0 | 0.40 | Yes 45° | 0.366 at 11.0GHz 0.313 at 15.0GHz 0.424 at 20.0GHz 0.48 at 24.0GHz 0.377 at 28.0GHz 0.309 at 30.0GHz |
| [37] | 60×60 4.60λo×4.60λo | FR4 1.60 | 14.0-18.0 | 08 | 25 | 0.008 | 9.96 | NC | NC | NO | NO |
| [38] | 70×70 0.66λo×0.66λo | FR4 1.60 | 3.03-15.33 | 08 | 10.0 | 0.10 | 9.9 | NC | NC | NO | NO |
| [39] | 8.5×22.5 1.21λo×3.20λo | FR4 0.50 | 26.0-28.0 | 08 | 20.0 | 0.005 | 9.97 | NC | NC | YES | 0.00427 at 27GHz |
| [40] | 70×70 1.20λo×1.20λo | FR4 1.60 | 3.15-6.0 | 08 | 10.0 | 0.0001 | 9.00 | NC | NC | NO | NP |
| [41] | 75×150 1.27λo×2.55λo | FR4 0.80 | 3.34-3.70 4.67-5.08 | 08 | 12.0 | 0.08 | NC | NC | NC | NO | NO |
| [42] | 54×54 5.30λo×5.30λo | Rogers5880 0.80 | 23.3-27.6 | 08 | 26.0 | 0.0004 | 9.98 | NC | NC | NO | NO |
| [44] | 90×90 0.97λo×0.97λo | FR4 1.524 | 2.84-15.3 | 08 | 20.0 | <0.015 | NC | <-11.0 | <0.30 | NO | NO |
| [45] | 38×90 0.55λo×1.30λo | Neltec 0.762 | 3.0-15.0 | 08 | 18.0 | <0.002 | NC | NC | <0.40 | NO | NO |
| [46] | 54×54 0.89λo×0.89λo | FR4 1.60 | 3.0-12.0 | 08 | 20.0 | <0.009 | >9.95 | <-10.0 | <0.35 | NO | NO |
| [47] | 85×85 1.40λo×1.40λo | FR4 0.80 | 3.0-10.4 | 08 | 15.0 | <0.02 | NC | <-10.0 | NC | NO | NO |
| [48] | 60×60 0.64λ0×0.64λ0 | FR4 1.60 | 3.20-5.32 | 08 | 15.0 | <0.13 | >9.90 | NC | NC | NO | NO |
| [49] | 75×150 1.04λ0×2.09λ0 | FR4 0.80 | 2.55-2.66 | 08 | 10.0 | <0.005 | NC | <-16.0 | NC | NO | NO |
| [52] | 54×54 5.24λ0×5.24λ0 | Rogers5880 0.80 | 23.3-27.60 | 08 | 26.0 | <0.0004 | >9.99 | NC | NC | NO | NO |
| [53] | 73.10×73.10 1.41λ0×1.41λ0 | FR4 1.60 | 3.52-4.00 | 08 | 16.0 | <0.0005 | ≅10.0 | <0.0 | <0.10 | NO | NO |

*(Continued)*

**Table 6.** (Continued)

| Ref. | Size of the Antenna (mm²) | Substrate/ Thickness (mm) | Bandwidth (GHz) | No. of Ports | Isolation | ECC | DG (dB) | TARC (dB) | CCL (b/s/Hz) | Bending Capability | SAR Analysis (W/Kg) |
|---|---|---|---|---|---|---|---|---|---|---|---|
| *P | 70.71 × 70.71 1.07λo × 1.07λo (≅5000mm²) | Rogers5880 0.254 | 3.59-3.85 8.40-70.0 | 08 | 15.0 | 0.10 | 9.9998 | 8.0 | 0.015 | 15° 30° 45° | 0.0331 at 3.50GHz 0.0435 at 10.0GHz 0.0131 at 15.0GHz 0.0209 at 24.0GHz 0.0171 at 26.0GHz 0.0116 at 28.0GHz 0.00334 at 38.0GHz 0.00336 at 39.0GHz 0.00296 at 41.0GHz 0.00149 at 47.0GHz 0.0851 at 60.0GHz |

NA-Not Applicable, NC-Not Calculated, *P-Proposed Eight-Port MIMO$_{n-s}$ Antenna.

## Supporting information

**S1 File: Data sheet.**

(XLSX)

## Author contributions

**Conceptualization:** Manish Sharma.

**Formal analysis:** Manish Sharma, Sathishkumar Nallusamy.

**Investigation:** Manish Sharma, Selvaraj Praveen Chakkravarthy, Rana Gill.

**Methodology:** Manish Sharma.

**Supervision:** Kanhaiya Sharma, Juliano Katrib.

**Validation:** Juliano Katrib.

**Writing – original draft:** Manish Sharma.

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
