## [Decision Letter · Decision Letter 0]

26 Feb 2025

PONE-D-24-59466Experimental Investigation of Flexible Eight-Port Asymmetric Fed MIMO Antenna with Narrow-Super-Widebandn-s Characteristics for Future Applications Including Internet of ThingsPLOS ONE

Dear Dr. Sharma,

Thank you for submitting your manuscript to PLOS ONE. After careful consideration, we feel that it has merit but does not fully meet PLOS ONE’s publication criteria as it currently stands. Therefore, we invite you to submit a revised version of the manuscript that addresses the points raised during the review process.

We look forward to receiving your revised manuscript.

Kind regards,

Sachin Kumar, Ph.D.

Academic Editor

PLOS ONE

Additional Editor Comments:

The article demonstrates the author's meticulous work in designing, developing, and analyzing an antenna for IoT-based wireless communication. The authors are requested to address the reviewers' comments and resubmit the manuscript for consideration.

Reviewers' comments:

Reviewer's Responses to Questions

**Comments to the Author**

1. Is the manuscript technically sound, and do the data support the conclusions?

Reviewer #1: Yes

Reviewer #2: Yes

2. Has the statistical analysis been performed appropriately and rigorously? 

Reviewer #1: Yes

Reviewer #2: Yes

3. Have the authors made all data underlying the findings in their manuscript fully available?

Reviewer #1: Yes

Reviewer #2: Yes

4. Is the manuscript presented in an intelligible fashion and written in standard English?

Reviewer #1: Yes

Reviewer #2: Yes

5. Review Comments to the Author

Reviewer #1: 1. Improve abstract. Add results statistics.

2. The introduction may be strengthened further.

3. Remove unnecessary items from the manuscript e.g. Fig.5, application band terminology, most common equations etc.

4. mention the importance of connected ground plane.

5. strengthen the conclusion.

6. incorporate latest research in the provided domain

Reviewer #2: The proposed work presents an eight-port MIMO antenna exhibiting super wideband characteristics for wearable applications. The antenna design methodologies have been explained and are supported by simulated and measured results. However, below are certain suggestions for further improvement of the proposed work:

1. A detailed explanation of the antenna design equations [1]-[4] could be included. An addition of this explanation will improve the reader's understanding of the design process.

2. Please include a justification for the selection of the resonant frequency, fr, as 20 GHz, considering that the antenna is super wideband.

3. The dimensions of the MIMO antenna (E1-E6) depicted in Fig. 6(a) and (b) could also be included in Table. 1.

4. Please consider fragmenting Fig. 6 into multiple figures for improved readability. The diagrams and fabricated prototype could be grouped in one figure, while the results could be presented separately.

5. The decoupling structure can be explained more explicitly highlighting its importance in achieving desired MIMO characteristics.

6. PLOS authors have the option to publish the peer review history of their article (what does this mean? ). If published, this will include your full peer review and any attached files.

**Do you want your identity to be public for this peer review?** For information about this choice, including consent withdrawal, please see our Privacy Policy .

Reviewer #1: No

Reviewer #2: **Yes: ** Dr. Chandni Bajaj

---

## [Author Response · Author response to Decision Letter 1]

6 Mar 2025

To,

The editor,

PLOS-ONE

Subject: Regarding submission of PONE-D-24-59466

The authors are thankful to reviewers for critically reviewing the manuscript and allowing revising the paper. The authors have addressed all the queries raised by both the reviewers and kindly find the point-to-point reply given below

Additional Editor Comments:

The article demonstrates the author's meticulous work in designing, developing, and analyzing an antenna for IoT-based wireless communication. The authors are requested to address the reviewers' comments and resubmit the manuscript for consideration.

The authors are thankful to the reviewer and Editor for providing the opportunity to revise the paper. The point-to-point reply of the reviewer(s) is given below in detail.

Reviewer #1:

1. Improve abstract. Add results statistics.

Ans. Authors are thankful to reviewer for addressing the content of the abstract and improving the same with statistical results. The abstract has been improved and same has been highlighted in the revised manuscript.

2. The introduction may be strengthened further.

Ans. Authors have further strengthened the introduction and respectively reference section by adding 12-port, 16-port MIMO antenna. Also, the relevant text has also been updated in the introduction and same has been highlighted in the revised manuscript.

3. Remove unnecessary items from the manuscript e.g. Fig.5, application band terminology, most common equations etc.

Ans. The objective of Figure 5 was to give more insight on MIMO terminology. As per your valuable suggestion, Figure 5 has been removed and also common equations from the revised manuscript.

4. Mention the importance of connected ground plane.

Ans. Authors are thankful to reviewer for raising the concern related to importance of commonly connected ground. The research article which is given below

[1] M.S. Sharawi, “Current misuses and Future Prospects for Printed Multiple-Input, Multiple-Output Antenna Systems,” IEEE Antennas & Propagation Magazine, pp. 162-170, 2017.

Cites that the unconnected ground isolates themselves from the other radiating element with no common current path available because it is not continuous. In real practice, the signals should have common ground so that all signal levels within the system can be interpreted properly based on that reference level (i.e., zero volts or GND level). If separate GND planes are used, one cannot guarantee that the system will work since the assumption of having all GND planes with the same voltage level is invalid. Thus, providing MIMO antenna configurations with multiple GND planes

should be avoided.

5. strengthen the conclusion.

Ans. The conclusion has been strengthened as per your valuable suggestions.

6. incorporate latest research in the provided domain

Ans. As per the suggestions, latest research papers have been added in the revised manuscript and same has been highlighted.

Reviewer #2:

The proposed work presents an eight-port MIMO antenna exhibiting super wideband characteristics for wearable applications. The antenna design methodologies have been explained and are supported by simulated and measured results. However, below are certain suggestions for further improvement of the proposed work:

1. A detailed explanation of the antenna design equations [1]-[4] could be included. An addition of this explanation will improve the reader's understanding of the design process.

Ans. Authors are thankful to reviewer for the significance of Equations 1-4. As per your suggestions, more explanation has been added in the revised manuscript and also the significance of fr=20.0GHz is mentioned in the revised manuscript and same has been highlighted.

2. Please include a justification for the selection of the resonant frequency, fr, as 20 GHz, considering that the antenna is super wideband.

Ans. The reason for selecting the fr=20.0GHz is due to the wider-impedance characteristics of the antenna, fr=20.0GHz is chosen and the corresponding iterations helps in improving the matching of the impedance around the center resonance frequency, fr GHz.

3. The dimensions of the MIMO antenna (E1-E6) depicted in Fig. 6(a) and (b) could also be included in Table. 1.

Ans. Authors are thankful for mentioning the dimensions concerns. The values of Fig. 6(a) and Fig. 6(b) has been tabulated in new Table 2 in revised manuscript and also same has been highlighted in the revised manuscript.

4. Please consider fragmenting Fig. 6 into multiple figures for improved readability. The diagrams and fabricated prototype could be grouped in one figure, while the results could be presented separately.

Ans. Authors are thankful to reviewer for suggesting the fragmenting of Fig. 6. The Fig. 6 is fragmented to Fig. 5 (due to the deletion of Fig. 5 which is suggested by Reviewer 1 in Q3) and Fig. 6 with Fig. 5 comprising of Simulation model and the fabricated prototype and Fig. 6 includes all the simulated-measured results.

5. The decoupling structure can be explained more explicitly highlighting its importance in achieving desired MIMO characteristics.

Ans. Role of Decoupling Structures in MIMO Antennas

In Multiple Input Multiple Output (MIMO) antenna systems, decoupling structures play a crucial role in enhancing performance by mitigating mutual coupling effects between closely spaced antenna elements. Here’s how they contribute:

1. Reduction of Mutual Coupling

• MIMO antennas are often designed with closely packed radiating elements to fit within compact devices.

• This proximity leads to strong electromagnetic (EM) coupling, causing signal interference and degrading isolation.

• Decoupling structures help suppress this coupling, ensuring that each antenna element transmits and receives signals independently.

2. Improved Isolation

• By minimizing mutual coupling, decoupling structures enhance the isolation between antenna elements.

• Improved isolation leads to better diversity gain, higher channel capacity, and reduced correlation, which are key factors in MIMO systems.

3. Enhanced Radiation Efficiency

• High mutual coupling distorts the radiation pattern of individual elements, leading to inefficiencies.

• Decoupling structures maintain proper radiation characteristics, improving overall efficiency.

4. Better Impedance Matching

• When antennas are coupled, their input impedance changes dynamically, causing impedance mismatch.

• Decoupling structures stabilize the impedance characteristics, allowing better matching and reduced reflection losses.

5. Increased Channel Capacity

• A MIMO system’s performance is dictated by how independently each antenna element can transmit and receive signals.

• Decoupling structures reduce correlation between antennas, leading to a higher channel capacity and improved data rates.

Fig 6(j) and Fig 6(k) shows the distribution of surface current with excited port1 for frequencies of 3.50GHz and 60.0GHz. As per the observations, the radiations radiated by Ant. A has no impact on the neighboring radiating antenna which is due to the de-coupling structure providing the additional path for the flow of current and the orthogonal arrangement of the pair of antennas as shown in Fig 5(a), Fig 5(b), Fig 5(c), and Fig 5(d).

We want to sincerely thank you for taking the time and effort to review my manuscript titled " Experimental Investigation of Flexible Eight-Port Asymmetric Fed MIMO Antenna with Narrow-Super-Widebandn-s Characteristics for Future Applications Including Internet of Things". I deeply appreciate your thoughtful and constructive feedback, which has been invaluable in improving the quality and clarity of my work.

Your expertise and insights have not only strengthened this manuscript but have also provided me with a deeper understanding of the subject. I truly value the time you dedicated to carefully evaluating the work and offering suggestions for improvement.

Once again, thank you for your support and contribution to this research.

---

## [Decision Letter · Decision Letter 1]

19 Mar 2025

Experimental Investigation of Flexible Eight-Port Asymmetric Fed MIMO Antenna with Narrow-Super-Widebandn-s Characteristics for Future Applications Including Internet of Things

PONE-D-24-59466R1

Dear Dr. Sharma,

We’re pleased to inform you that your manuscript has been judged scientifically suitable for publication and will be formally accepted for publication once it meets all outstanding technical requirements.

Kind regards,

Sachin Kumar, Ph.D.

Academic Editor

PLOS ONE

Additional Editor Comments (optional):

The authors properly addressed the comments of the reviewers, and the manuscript is accepted for publication.

Reviewers' comments:

Reviewer's Responses to Questions

**Comments to the Author**

1. If the authors have adequately addressed your comments raised in a previous round of review and you feel that this manuscript is now acceptable for publication, you may indicate that here to bypass the “Comments to the Author” section, enter your conflict of interest statement in the “Confidential to Editor” section, and submit your "Accept" recommendation.

Reviewer #1: All comments have been addressed

Reviewer #2: All comments have been addressed

2. Is the manuscript technically sound, and do the data support the conclusions?

Reviewer #1: Yes

Reviewer #2: Yes

3. Has the statistical analysis been performed appropriately and rigorously? 

Reviewer #1: N/A

Reviewer #2: Yes

4. Have the authors made all data underlying the findings in their manuscript fully available?

Reviewer #1: Yes

Reviewer #2: Yes

5. Is the manuscript presented in an intelligible fashion and written in standard English?

Reviewer #1: Yes

Reviewer #2: Yes

6. Review Comments to the Author

Reviewer #1: The comments are addressed adequately. The paper has addressed all the queries.

 ..............................................

Reviewer #2: The authors have addressed all my queries satisfactorily, and the revisions have significantly improved the quality of the manuscript. In my opinion, the paper is now suitable for publication.

7. PLOS authors have the option to publish the peer review history of their article (what does this mean? ). If published, this will include your full peer review and any attached files.

**Do you want your identity to be public for this peer review?** For information about this choice, including consent withdrawal, please see our Privacy Policy .

Reviewer #1: No

Reviewer #2: **Yes: ** Dr. Chandni Bajaj

---

## [Editor Report · Acceptance letter]

PONE-D-24-59466R1

PLOS ONE

Dear Dr. Sharma,

I'm pleased to inform you that your manuscript has been deemed suitable for publication in PLOS ONE. Congratulations! Your manuscript is now being handed over to our production team.

Kind regards,

on behalf of

Dr. Sachin Kumar

Academic Editor

PLOS ONE